# OpenReview forum: "Optimization Proxies using Limited Labeled Data and Training Time - A Semi-Supervised Bayesian Neural Network Approach"
_ICLR.cc/2025/Conference — Submitted to ICLR 2025_

### Official Review · Reviewer_MUif · 2024-10-27

**Soundness:** 2
**Presentation:** 1
**Contribution:** 2
**Rating:** 6
**Confidence:** 3

**Summary:**

The authors introduces a semi-supervised Bayesian Neural Network (BNN) framework to efficiently solve constrained optimization problems common in engineering applications like energy networks, where uncertain parameters and limited labeled data pose computational challenges. By alternating between supervised cost-minimization with labeled data and unsupervised constraint satisfaction with unlabeled data, the model achieves a tenfold reduction in equality gaps and halves feasibility and optimality gaps compared to standard BNNs and deep neural networks. Additionally, a novel Selection via Posterior (SvP) scheme leverages BNN uncertainty estimates to further minimize errors, while the framework’s probabilistic confidence bounds offer a scalable solution adaptable to various high-stakes optimization problems with minimal labeled data requirements.

**Strengths:**

The strengths of this paper lie in its development of a semi-supervised Bayesian Neural Network (BNN) approach that addresses constrained optimization challenges under strict time and data constraints. First, the choice of BNNs over DNNs enhances uncertainty quantification, facilitating more reliable predictions by integrating prior beliefs, and is novel. The introduction of a novel Sandwich learning method further strengthens the model by enforcing feasibility through unlabeled data, bypassing the need for additional labeled instances. Moreover, the use of predictive variance within the BNN framework enables the formulation of tight expected error bounds.

**Weaknesses:**

- The authors should correct the citation formatting throughout the paper, particularly the use of \citet and \citep. For example, the citation of Ibrahim et al. (2020) should be formatted as (Ibrahim et al. 2020) rather than including it in the narrative.
- The overall presentation of the study is weak and requires enhancement for improved clarity and impact.
- A comparison against methods that require more training time is necessary. Since the neural networks proposed by Park & Van Hentenryck (2023) and others do not necessitate extensive training time, it is crucial to understand the implications of low computational requirements. Although low compute restrictions during test time are typically essential for inference on new examples, we need to train fewer models and thus it may render such restrictions (e.g., a hard limit of 10 minutes of training time on a single CPU core) unnecessary.
- What is the justification for including equation 1d as part of the constraints?
- The assertion that constructing the feasibility dataset Df​ incurs no additional computational cost should be revisited. While input sampling may indeed be inexpensive, obtaining feasible solutions can be resource-intensive, necessitating a more accurate representation.
- The authors should clarify why they did not compare their approach with the primal-dual self-supervised learning method proposed by Park & Van Hentenryck (2023). Additionally, the methods of DC3, and Zamzam & Baker (2020) should be included in the comparison. A new table could have been created to compare the training times of these methods, as it is possible to obtain outputs from these models without fitting them completely.
- Completely omitting state-of-the-art baselines does not appear to be a prudent choice. The authors should include the results of these methods while indicating that they require more time and thus cannot be used for direct comparison.
- Additionally, the authors have not accounted for the supervised dataset creation time within the 10-minute limit. Since self-supervised methods do not necessitate this step, the current comparison may be considered unfair. The authors should revise the training time to reflect a total of 10 minutes for training plus T minutes for generating true labels for the supervised dataset. This revised training time should then be applied uniformly to both the supervised and self-supervised methods, even when the latter do not require true labels.
- Is the statement, “Moreover, unsupervised methods still require testing data and consequently the associated data generation time in order to perform validation and provide confidence bounds on error with respect to true solution,” not applicable to the proposed method? Although the proposed method does not require data generation for confidence bounds, it does require labeled training data, which is not needed for self-supervised methods. Additionally, to compute the gap and constraint violations, wouldn’t the proposed method also need true labels?
- The manuscript contains several grammatical mistakes that should be addressed, including but not limited to:
    - "The fundamental idea of this training method is to update the network weights and biases through multiple rounds of training in which each round alternates between using the labeled dataset."
    - "Next, we present results for Probabilistic Confidence bounds, described in Section 4. Figure 3 shows that..."
    - "In Bayesian Neural Network (BNN) literature, the standard approach is to use the mean posterior prediction Ep​(w∣D)\[fw​(xt​)] for a test input xt​." Please clarify the context for "a standard approach."

**Questions:**

Please refer to the section on weaknesses.

---

> ### Author Response · Authors · 2024-11-23
> **Reply to Reviewer MUif**
>
> We thank the reviewer for their comments.
>
> 1. We have used the bibliographystyle\{iclr2025\_conference\} for citations and will re-check and update the reference format as per guidelines.
> 2. We would like to request the reviewer to expand more on this to specify what improvements can be made by the authors for the reviewer to adjust the score. Additionally, we have decided to update the manuscript with one motivation diagram and some additional results on different dataset comparing with other state of art methods like DC3. Detailed results, to be updated in the revised manuscript, can be found on: https://drive.google.com/file/d/1Y57JVPegi2HY2krnLb7qihMvQNdWvfdg/view?usp=share_link
> 3. Our focus was on ensuring feasibility without compromising optimality in settings constrained by limited training data and time. The goal was to develop an optimization proxy for practical yet challenging scenarios where labeled data and total training and testing times are restricted (see Figure 1: (https://drive.google.com/file/d/1Y57JVPegi2HY2krnLb7qihMvQNdWvfdg/view?usp=share_link)).  This differs from scenarios with abundant training samples or extended training ($T_{\text{training}}$) and testing ($T_{\text{prediction}}$) times. Notably, $T_{\text{prediction}}$ for our model is at least **10 times lower** than approaches like DC3 or Zumzum & Baker's, as it avoids projection steps requiring power flow solvers during training or testing. Additionally, our results show up to an order of magnitude improvement in feasibility gaps without compromising optimality. For example, the optimality gap for case118 reduces to **0.089** with our method, compared to **1.284** for the best point prediction method (Table 2). Our BNN-based proxy also generates meaningful confidence bounds on prediction accuracy despite using only 1,000 validation samples. As shown in Figure 4, prior DNN-based models, even with the same validation data and using Hoeffding's or Empirical Bernstein inequalities, fail to provide meaningful bounds.  Self-supervised methods like primal-dual learning (Park & Van Hentenryck, 2023) require significantly longer training times, e.g., **5,932.5 seconds** for the 57-bus system and **7,605.1 seconds** for the 118-bus system. In contrast, our method trains within **600 seconds**, making their training times 10 times longer. For this reason, we did not use such methods for benchmarking.
> 4. We included 1d to be explicit about the input to the optimization problem.  We will move it to the paragraph below equation (1)  in the revised manuscript.
> 5. We respectfully disagree with the reviewer’s concern regarding the computational intensiveness of constructing the feasibility dataset $D_f$. The process of creating $D_f$ is, in fact, computationally light because it relies solely on the necessary conditions for feasibility in constrained optimization problems. For any feasible solution to a constrained optimization problem, the constraints must be satisfied exactly—resulting in zero constraint violations or "gap" as defined in equation (2). Thus, by definition, the feasibility dataset $D_f$ is constructed such that each entry in $D_f $ corresponds to a constraint satisfaction value of zero. Specifically, given any input $ \mathbf{x}$ in $ D_f $, the output is identically zero, indicating that all constraints are met. Therefore, constructing $ D_f $ is as computationally inexpensive as simply sampling inputs, since there is no additional cost for evaluating or solving complex constraint satisfaction conditions. The feasibility requirement simplifies the dataset creation process, ensuring that the computational load remains minimal. This part is mentioned in paragraph below equation (2) of the manuscript.
> 6. As mentioned before, the proposed works motivation is to learn an ACOPF proxy under limited training data and training time situations. Therefore, methods which takes considerably large amount of time to train (approx. 6000 sec. for by Park & Van Hentenryck (2023) for 57-Bus system) does not fit within the motivation of this work. Moreover, an important difference between methods such as DC3 and  Zamzam & Baker (2020) is presence of power flow in the pipeline during training and prediction. The inclusion of power flow implies that we need to solve nonlinear equations within the proxy to obtain accurate predictions. This leads to considerable increase in both training and prediction time. For example, in DC3 paper it is listed that it takes 0.089 sec. to predict the ACOPF solution for one instance, which is approximately 30 times higher than 0.003 sec prediction or inference time of the proposed method. This limits the use of use of these models in stochastic settings such as probabilistic risk quantification where a large number of (order of millions) predictions are needed.

---

> ### Author Response · Authors · 2024-11-23
> **Reply to Reviewer MUif (Point 7 onwards)**
>
> 7. We would like to highlight that the manuscript includes five different state-of-art baselines, for all the systems. Thus, we disagree that baselines were omitted. Also, as noted in the paper and before, self-supervised methods are very time consuming in training (Authors in Park & Van Hentenryck (2023) report that **5932.5 seconds** are taken to train models on 57-bus system and **7605.1 seconds** are taken to train on 118-bus systems.) The primary motivation of this paper is to develop an Optimization proxy under a practical but challenging situation where total labeled data for training as well as total time for training and testing are constrained. Moreover, to strengthen the discussion, we will update the comparative results in the revised manuscript, which can be found here: https://drive.google.com/file/d/1Y57JVPegi2HY2krnLb7qihMvQNdWvfdg/view?usp=share_link
> 8. Yes we agree that we have not accounted for time for generating 512 training samples. However, we do define the total time to be data generation + training time, in which data generation time is constant for all different models. We have not commented on data generation time as we have used open source **torch_geometric** dataset which does not provide generation time. However, in the revised manuscript, we will update the training data generation time using **PowerModels.jl** package. It takes on average 0.15 sec. to solve an instance of ACOPF for 57-Bus and 0.45 sec. for 118-Bus system ACOPF. Therefore, it takes 45.36 sec to generate training + validation set for case57 and 136.08 for case118, on a five core CPU. In practice, this data is can also be available from historical operations. Further, it is clear that these data generation time is much slower than training time of 600 sec. opted in this work.
> 9. Yes, the proposed method requires validation samples to generate the gap and constraint violation. The statement was written to highlight that unsupervised methods are not completely free from labeled data requirements. We understand that this might be misleading, hence we will qualify the statement as follows: ``Moreover, unsupervised methods, similar to supervised and proposed method also require validation data and consequently incur data generation time. This validation is needed to provide confidence bounds on error with respect to true solution." However, we want to highlight that for the same set of validation data, DNN based models produce trivial confidence bounds due to the use of only Hoeffding's inequality as highlighted in section 4 of the paper. Our method is able to leverage the sampling of weights inside the BNN to obtain better confidence bounds through the use of Bernstein inequality.
>
> On Mistakes:
>
> We thank the reviewer for careful reading. We will proof read and revise the manuscript.
>
> A standard approach in the context of of mean prediction is written to reflect that we do not opt for weighted mean approach, instead go with most used approach of exception calculation for mean posterior. We will qualify the statements to reflect the same in revised manuscript.

---

> > ### Comment · Reviewer_MUif · 2024-11-27
> >
> > Thank you for your response. I have updated my score accordingly.
> >
> > However, I remain unconvinced regarding the constraints of limited labeled data and restricted training time. Recent advances in self-supervised (unsupervised) learning for constrained optimization *(e.g., Park and Hentenryck, 2023; Arya, Rahman, and Gogate, 2024)* address these challenges effectively, as they do not rely on labeled data. In contrast, supervised methods require the generation of optimal solutions for true labels, and the model’s performance is inherently tied to the quality of these labels. This can introduce additional bottlenecks, particularly in scenarios where high-quality labels are expensive or difficult to obtain.
> >
> > Reducing inference time is critical, particularly for real-time applications. However, even under a restricted training time constraint, self-supervised models could still be trained within the available time (**even if not until convergence**) and meaningfully compared with the proposed methods. Furthermore, since self-supervised methods do not require the additional time for label generation, the training time could be reallocated to parameter learning, potentially resulting in improved model performance.
> >
> > Finally, I do not find the constraint of limited labeled training data particularly compelling, as self-supervised methods (Park and Hentenryck, 2023; Arya, Rahman, and Gogate, 2024) do not require labeled data, and, as the authors noted, sampling data is not particularly time-intensive. This suggests that the limited training data constraint could be circumvented by adopting self-supervised approaches.
> >
> > That said, I appreciate the novelty of incorporating Bayesian semi-supervised learning, the sandwich BNN framework, and probabilistic confidence bounds, which are significant contributions.
> >
> >
> >
> > Park, S. and Hentenryck, P.V. (2023) ‘Self-Supervised Primal-Dual Learning for Constrained Optimization’.
> >
> > Arya, S., Rahman, T. and Gogate, V. (2024b) ‘Learning to Solve the Constrained Most Probable Explanation Task in Probabilistic Graphical Models’, in Proceedings of The 27th International Conference on Artificial Intelligence and Statistics.PMLR, pp. 2791–2799.

---

> > > ### Author Response · Authors · 2024-11-28
> > > **Reply to Reviewer MUif**
> > >
> > > We thank the reviewer for their constructive feedback and updated score. We have the following comments related to the remaining queries:
> > >
> > > In the context of the labeled data discussion, we want to highlight that, similar to the proposed and supervised learning methods, the unsupervised methods, like proposed by Park & Pascal (2023) also need **validation datasets**. Now, by extending the reviewer’s argument, let us consider a situation where an unsupervised learning approach is used, and 1512 samples (same as the total training + testing samples in the proposed work) are available for validation and error bounding. Using Hoeffding's bound with $R=1$ and $\delta = 0.05$, one obtains:
> > > $$
> > > \varepsilon = \frac{0.9}{\sqrt{1512}} \approx 0.02.
> > > $$
> > > Clearly, this is a very loose bound, as voltage value variations are typically of the order of $10^{-2}$.
> > >
> > > Now, one can use Empirical Bernstein bounds, leveraging the validation samples to calculate the empirical total variance in error $\widehat{\mathbb{V}}_e$ using all 1512 samples. In this case, the error bound will be:
> > > $$
> > > \varepsilon = \frac{1.88\sqrt{\widehat{\mathbb{V}}_e}}{\sqrt{1512}} + \frac{5.3344}{1512} = 0.048\sqrt{\widehat{\mathbb{V}}_e} + 0.0035.
> > > $$
> > >
> > > Here, we want to emphasize that **although Empirical Bernstein bounds have been available in the literature for a long time, they are not commonly used for error-bounding exercises in works related to constrained optimization proxies, neither in supervised nor in unsupervised settings. Thus, one minor yet important contribution of our work is the reintroduction and contextualization of Empirical Bernstein-based probabilistic bounds in optimization proxy settings.**
> > >
> > > Now, let us compare this with the proposed work's setting, where only 1000 samples are available for validation, along with the hypothesis that $2\text{MPV}$ is an upper bound of the total variance in error and that the second term of the total variance in error is small. (Note that this hypothesis has been stated in the paper, discussed intuitively, and validated with various test cases.) Using the theoretical Bernstein bounds—**which cannot be used by unsupervised methods as MPV information is only available with BNNs**—we obtain:
> > > $$
> > > \varepsilon = \frac{2.88\sqrt{\text{MPV}}}{\sqrt{1000}} + \frac{0.867}{1000} = 0.091\sqrt{\text{MPV}} + 0.0008.
> > > $$
> > >
> > > By analyzing Empirical and Theoretical Bernstein-based error values, with 1512 and 1000 validation samples for unsupervised and proposed learning methods respectively, we make the following assertions:
> > >
> > > **1.** The second term of the proposed method’s bound is significantly smaller, $0.0008 < 0.0035$, compared to what one can achieve with unsupervised methods.
> > >
> > > **2.** Once the proposed method achieves MPV values on the order of $10^{-3}$ or lower, the proposed Theoretical Bernstein bound will always be tighter than the Empirical Bernstein bounds achieved by unsupervised methods, even if $\widehat{\mathbb{V}}_e$ is on the order of $10^{-4}$ —an order of magnitude better than the proposed method.
> > >
> > > In view of the extensive testing performed in the paper, we argue that the proposed method can achieve tighter bounds with the same number of total labeled data samples, even if unsupervised methods converge to better solutions. We also want to highlight again that the 10 minutes of training time in the proposed method is very short compared to the 120 minutes required by the unsupervised method presented by Park and Pascal (2023).

---

### Official Review · Reviewer_KQjj · 2024-11-01

**Soundness:** 2
**Presentation:** 1
**Contribution:** 2
**Rating:** 3
**Confidence:** 4

**Summary:**

The paper introduces a semi-supervised learning framework based on BNN to solve constrained optimization problems when labeled data is limited and model training times are restricted.
The authors propose a sandwich-style training approach alternating between supervised and unsupervised learning. Additionally, they employ BNNs to generate multiple predictions and improve feasibility through a selection process. Experimental results demonstrate that BNNs outperform DNNs in settings with low data and limited computational resources.

**Strengths:**

1. The work considers a practical and important scenario of solving constrained optimization problems when labeled data is limited and model training times are restricted.
2. The authors derive tight confidence bounds for the testing error by utilizing Bernstein's inequality.
3. The experiments conducted on the case 2000 demonstrate the scalability of the proposed approach.

**Weaknesses:**

1. While the main challenges addressed in this work are limited labeled data and restricted model training times, it is unclear how BNNs specifically contribute to addressing these challenges. The sandwich-style semi-supervised training approach proposed by the authors can be applied to any NN structure.
    - Training BNNs is computationally more expensive than regular NNs. The authors should discuss it and present the complexity explicitly.
    - In section 3.1., the authors propose a selection via posterior strategy to reduce the constraint violation. It is unclear what the benefits of BNN are in this strategy since one can add Gaussian noise to regular DNN prediction and select the one with minimum constraint violation. Theoretically, how does the equality constraint violation decrease with the increased generated samples?
    - In Section 4, the authors derive confidence bound for the testing error using the MPV as a proxy for the TVE in the Bernstein bound. They hypothesize an inequality (line 286) without providing sufficient justification. The authors should explicitly state and discuss the assumptions made in deriving this confidence bound.
    - The authors should discuss and compare their confidence bound with existing generalization analyses or confidence bounds for DNNs based on the number of training samples, such as those presented in [1].

2. The authors propose an alternated training approach where supervised and unsupervised learning are performed in separate iterations. However, the benefits of this approach are unclear. Since one can easily combine the supervised and unsupervised loss together at each iteration, it may save more training time.

3. Overall, this work combines sandwich training, BNN, and SvP, so a comprehensive ablation study is needed to evaluate the individual contributions and importance of each component.

4. In experiments, the authors only include supervised training methods as baselines, excluding self-supervised and primal-dual learning approaches due to their higher training times and computational demands.
    - The sandwich training approach can also be applied to regular NNs, serving as a meaningful baseline.
    - There are efficient unsupervised approaches for solving AC-OPF problems that the authors did not discuss in the related work section or compare against in their experiments [2].

[1] Kawaguchi, K., Kaelbling, L. P., & Bengio, Y. (2017). Generalization in deep learning. arXiv preprint arXiv:1710.05468, 1(8).

[2] Huang, W., & Chen, M. (2021). DeepOPF-NGT: Fast no ground truth deep learning-based approach for AC-OPF problems. In ICML 2021 Workshop Tackling Climate Change with Machine Learning.


I will adjust my scores if the concerns are addressed.

**Questions:**

1. How does the number of labeled or unlabeled training data affect the BNN performance?
2. How does the number of posterior samples affect the constraint violation?

---

> ### Author Response · Authors · 2024-11-23
> **Reply to Reviewer KQjj**
>
> We thank the reviewer for their comments.
>
> Point 1:  Reviewer is correct to say that semi-supervised training approach proposed in this work can be applied to any NN. However, we want to argue that with limited training data, A BNN is better suited for learning ACOPF than NNs and DNNs. This argument can be corroborated from results presented in the paper where simple BNN always outperforms a NN with both MSE and MAE loss function (Comparative result Table 1 and 2 in Section 5 and  Table 6 and 7 in Appendix C). Moreover, when using NN we do not have access to predictive uncertainty and thus cannot use the superior theoretical Bernstein inequality for error bounds.
>
> -- Yes, training BNN is more expensive than regular NN. However, in the present works setting, the training is constrained by the time and number of data points available are very limited. In such settings, the BNN outperforms NNs as shown in the results. We have adopted a setting motivated by  practical considerations where there is a hard upper bound on the total time available. This is a different setting where the theoretical complexity of training may not be as relevant compared to what is the best that can be done in limited time. Please find details of motivation and additional results here: https://drive.google.com/file/d/1Y57JVPegi2HY2krnLb7qihMvQNdWvfdg/view?usp=share_link
>
> -- Firstly, we want to highlight that for  selection via posterior strategy, reduction of the constraint violation is only one of the criteria. One can easily design a similar strategy  for minimum cost, minimum feasibility error or a weighted combination of both. BNNs provide a principled way to quantify the uncertainty in the predicted output, which is not the same as adding arbitrary noise to NNs. In BNNs, the weights are probabilistic and each weight of the network has a non trivial posterior distribution which is updated during training. The distribution of the output is then a consequence of the weight distribution. The significantly larger probabilistic search space afforded by the weight distributions (often bimodal in our case) can lead to an output distribution that very different from Gaussian, even when the weight distributions started with a Gaussian prior. Therefore, using Gaussian noise to the output in NNs will not correspond it the same thing. For uncertainty quantification through NNs, a prominent approach is to use Ensemble of NNs, which provide output distribution via multiple possible weights. However, training large number of NNs require a lot of training time+ resource along with variations in training dataset. Thus, in the current works limited time and data setting it won't be feasible and BNN based approach is the preferred method both for accuracy and confidence bound generation.
>
> -- The hypothesis of using MPV as a proxy for TVE is motivated by our empirical observations. For ACOPF proxies, we observed that two times MPV is always greater than the total variance in error. Furthermore, as explained from line 294 onward, the first term of the total variance in error is the same as MPV. If the second term of the total variance (expectation over weights and variance over samples) is lower than MPV, our hypothesis holds. For ACOPF, we observed that the second term of TVE (Equation 5) is significantly lower than MPV (the first term of TVE). Figure 3 has been included to validate the hypothesis for various power system test cases. Note that Figure 3 shows results for both supervised and sandwich BNNs, covering a total of six instances where the hypothesis holds.We have also observed the same characteristic of MPV in other test cases. We will clarify and update the hypothesis discussion in the revised manuscript.
>
> -- Our confidence bound is similar to Proposition 5 in [1]. In the premise of Proposition 5, an absolute bound on the error (C) and on the second moment ($\gamma^2$) is assumed and the result is derived using the theoretical Bernstein inequality. In our setting, by using the automatically generated MPV from the training phase of BNNs along with hypothesis in line 286 or Eq 5 (clarified in our response to the previous question) we are able to obtain a similar confidence bound. Note that in either case, this bound is to help quantify the accuracy over i.i.d. out of sample test data. Section 4 in our paper is not a attempting to generalize the training accuracy, that is complicated due to dependence between model and the training samples. We will cite [1] in the revised document.

---

> ### Author Response · Authors · 2024-11-23
> **Reply to Reviewer KQjj (Point 2 Onwards)**
>
> Point 2: We want to clarify that one cannot combine supervised and unsupervised losses in Bayesian Neural Networks (BNNs) without having a tight lower bound on the cost function with unlabeled input data. This is because BNNs are trained using the Bayesian inference principle, where a likelihood function needs to be defined for the output. In the unsupervised learning stage, the proposed method relies on the fact that the observation for the feasibility gap will be zero for any input. This allows the definition of a likelihood for constraint satisfaction that attempts to achieve a distribution centered around zero (although a perfect delta distribution is not achievable due to numerical issues). To construct a similar likelihood for the cost  (as done in the supervised training phase), we require tight bounds on the cost for unlabeled data.
>
> The setting of traditional NNs is different, as one can define a loss function that combines cost and weighted feasibility terms. These NNs are often referred to as Physics-Informed Neural Networks (PINNs). We have compared our proposed approach with these models (MAE + penalty and MSE + penalty) across various test cases presented in the paper. The results show that the proposed BNN approach outperforms these PINNs.
>
> Point 3: The section 5 of the manuscript provide separate contribution of each of the components. For example, first row of table 1 shows improvement via Sandwich BNN SvP compared to the Sandwich BNN where both cost function and feasibility gap improves. Similarly, comparing the Sandwich BNN and supervised BNN results indicate benefits of Sandwiching.
>
> We will update the discussion on the revised paper to highlight contribution of each part separately.
>
> Point 4: As mentioned in the manuscript, the proposed work's motivation is to learn an ACOPF proxy under limited training data and training time situations. Therefore, methods which takes considerably large amount of time to train (approx. 6000 sec. for by Park & Van Hentenryck (2023) for 57-Bus system) does not fit within the motivation of this work. Moreover, an important difference between proposed approach and methods such as DC3 and  Zamzam & Baker (2020) is presence of power flow in the pipeline during training and prediction. The inclusion of power flow implies that we need to solve nonlinear equations within the proxy to obtain accurate predictions. This leads to considerable increase in both training and prediction time. For example, in DC3 paper it is listed that it takes 0.089 sec. to predict the ACOPF solution for one instance, which is approximately 30 times higher than 0.003 sec prediction or inference time of the proposed method (with 100 weight samples). This limits the use of use of these models in stochastic settings such as probabilistic risk quantification where a large number of (order of millions) predictions are needed.
>
> Detailed results, to be updated in the revised manuscript, can be found on : https://drive.google.com/file/d/1Y57JVPegi2HY2krnLb7qihMvQNdWvfdg/view?usp=share_link
>
> Similar to methods such as DC3 and  Zamzam & Baker (2020), the work Huang, W., & Chen, M. (2021) also have power flow in the pipeline during prediction. The inclusion of power flow implies that we need to solve nonlinear equations within the proxy to obtain accurate predictions. This leads to considerable increase in both training and prediction time. For example, in DC3 paper it is listed that it takes 0.089 sec. to predict the ACOPF solution for one instance, which is approximately 30 times higher than 0.003 sec prediction or inference time of the proposed method (with 100 weight samples). This limits the use of use of these models in stochastic settings such as probabilistic risk quantification where a large number of (order of millions) predictions are needed.
>
> Questions:
>
> 1. We have observed that the proposed method also follows the general principle where the prediction error decreases with increase in training data. However, when the training time is limited, increasing the number of training samples beyond a certain threshold does not provide additional improvements. In our experiments, we observed that for systems larger than 118-Bus, increasing training samples beyond 1000 does not provide further benefits when the training time is limited to 600 sec. We will append these results to the revised manuscript.
>
> 2. The effect of number of posterior samples on output is presented in the Figure 5, Appendix C. It shows that errors in various parameters stabilizes around 200 posterior samples.

---

> > ### Comment · Reviewer_KQjj · 2024-11-24
> > **Response to rebuttal**
> >
> > Thank you for your detailed response. While I acknowledge your efforts to address the concerns, I still have some important points to discuss regarding the BNN methodology and experimental validation:
> >
> > Regarding Experimental Design:
> > - While the author mentioned that they compared the proposed approach and baseline models (MAE + penalty and MSE + penalty), it seems an **unfair** comparison. Specifically, the supervised baselines utilize only labeled data for MAE/MSE and penalty calculations, but the sandwich/semi-supervised training leverages both labeled and unlabeled data. So, it is not surprising that BNN has smaller constraint violations. One straightforward and fair baseline is using sandwich/semi-supervised training for regular DNN with (the same) labeled and unlabelled data, as I mentioned in my initial review.
> > - Regarding the computational concerns with power flow equations, I would like to highlight two established approaches from the literature:
> >   - Predicting voltage and power generation, with remaining variables solved via Newton's method (e.g., Donti. (2021.)). While computationally intensive, this ensures strict feasibility.
> >   - Predicting voltage magnitude and angle, with power generation solved through closed-form calculations (e.g., Huang, W., & Chen, M. (2021)). This offers negligible computational overhead but has a power flow mismatch.
> >
> >   Therefore, as suggested in my initial review, the author should discuss and consider the unsupervised training approach as a baseline, which does not incur significant computational issues from the power flow calculation.
> >
> > Regarding the motivation of BNN:
> > - The current justification for BNN usage relies primarily on experimental results. Given the aforementioned concerns about experimental comparisons, a stronger theoretical/emprirical justification is needed.
> > - Specifically: What unique theoretical advantages does BNN offer over DNN for learning **deterministic** mappings?

---

> > > ### Author Response · Authors · 2024-11-25
> > > **Reply to Reviewer KQjj on Experimental Results**
> > >
> > > We would like to highlight three key points in light of the **experimental results**:
> > >
> > > 1. **Superiority of Supervised BNNs**
> > >    The experimental results clearly show that a simple supervised BNN significantly outperforms DNN models. We would like to emphasize that our contribution includes using BNNs to create ACOPF proxies, not just the sandwich learning model. Based on the results comparing supervised BNNs and supervised DNNs, we argue that BNNs are more suitable for learning proxies under low training data settings. This empirical, results-based assertion aligns with findings in various BNN studies [R1-R2].
> > >
> > > 2. **Cost of Unsupervised Training**
> > >    We want to stress that unsupervised training is computationally expensive. While some papers do not explicitly report training times, the state-of-the-art unsupervised learning work [Park and Pascal] indicates that primal-dual learning (Park & Van Hentenryck, 2023) requires significantly longer training times—for example, **5,932.5 seconds** for the 57-bus system and **7,605.1 seconds** for the 118-bus system. In contrast, our method trains within **600 seconds**, making their training times approximately 10 times longer. For this reason, we did not use such methods for benchmarking.
> > >
> > > 3. **Projection Using Power Flow**
> > >    Achieving a zero equality gap requires solving nonlinear power flow equations, as done in DC3 and by Zumzum & Baker. However, the method used by Huang, W., & Chen, M. (2021) does not achieve a zero gap. After predicting voltage and angle, the $P_g$ and $Q_g$ values (obtained from equations (2) and (3)) are fixed, leaving no degrees of freedom to satisfy nodal power balance. This limitation is reflected in their results, where load satisfaction is less than 100%. However, the nodal balance violation (i.e., equality gap) information is not explicitly provided.
> > >
> > >    Additionally, the equation-solving method proposed by Huang, W., & Chen, M. (2021) is directly applicable to our proposed method. It could even enable an SvP-style mechanism to derive optimal weights through linear equation solving.
> > >
> > > Lastly, while we agree that a sandwich DNN could serve as a baseline for BNN models, it would constitute a distinct contribution. This is because, with DNNs, it is crucial to explore merging supervised and unsupervised layers, which is not feasible with BNNs, as highlighted in our earlier response.
> > >
> > > [R1] Laurent Valentin Jospin et. al.,
> > > "Hands-on Bayesian neural networks—A tutorial for deep learning users." IEEE Computational Intelligence Magazine 17, no. 2 (2022): 29-48
> > >
> > > [R2]Stefan Depeweg et. al. "Decomposition of Uncertainty in Bayesian Deep Learning for Efficient and Risk-sensitive Learning", Proceedings of the 35th International Conference on Machine Learning, PMLR 80:1184-1193, 2018.

---

> ### Author Response · Authors · 2024-11-25
> **Reply to Reviewer KQjj : On Motivation**
>
> The key advantage that BNNs—or Bayesian methods in general—offer over DNNs and other deterministic methods in any learning problem (not just optimal power flow) lies in their performance under limited data conditions. With a sufficiently large dataset, DNNs or other universal approximation methods should perform equally well, if not better. However, as stated in the Introduction, the realistic regime for power grid optimization problems involves limited data and constrained training time.  Another perspective when comparing BNNs and DNNs is that BNNs can separate two types of uncertainty **Epistemic uncertainty**, which reflects the uncertainty in model parameters ($p(w|D)$, where $w$ represents the parameters and $D$ is the training data).  **Aleatoric uncertainty**, which represents the inherent uncertainty in the data itself ($p(y|x, w)$, where $y$ is the output and $x$ is the input).  This separation allows BNNs to be highly data-efficient, enabling them to effectively learn from small datasets without overfitting [R1–R2]. During prediction, instead of confidently producing an incorrect result, BNNs assign high epistemic uncertainty to points far from the training dataset, signaling that the model lacks sufficient knowledge about them. This eliminates the need to rely on **Ensemble DNNs** to account for weight distributions and build model robustness. Consequently, BNNs save significant computational resources that would otherwise be required to train multiple DNN models to create an ensemble.
>
> Additionally, we have demonstrated that the inherent randomness in BNNs allows for the rapid development of much better confidence bounds on predicted outputs compared to DNNs on similarly sized test datasets. This is primarily because the mean predicted variance of a BNN better approximates the true variance of outputs than the empirical variance computed by DNN models. While this observation is still empirical, our results confirm its validity across a range of power grid networks, including large networks that have not been addressed in existing DNN-based studies in this domain. Assuming that the variance predictions are reliable, we show that our model reduces confidence error scaling from $1/\sqrt{M}$ to $1/M$, where $M$ is the number of test samples.
>
> Finally, we argue that the proposed work offers a novel perspective on ACOPF and optimization proxies. The ability of BNNs to quantify uncertainty in predictions facilitates effective active learning and Bayesian Optimization [R3]. This work lays the foundation for using BNNs as surrogates in various engineering problems, where active learning and/or Bayesian Optimization can be employed to sample more informative datasets. Therefore, this study should not only be evaluated from a state-of-the-art standpoint but also as a contribution that opens new directions in the field.
>
>
> **We will ensure that the manuscript is updated with these discussions and additional details.**
>
> [R1] Laurent Valentin Jospin et. al.,
> "Hands-on Bayesian neural networks—A tutorial for deep learning users." IEEE Computational Intelligence Magazine 17, no. 2 (2022): 29-48
>
> [R2]Stefan Depeweg et. al. "Decomposition of Uncertainty in Bayesian Deep Learning for Efficient and Risk-sensitive Learning", Proceedings of the 35th International Conference on Machine Learning, PMLR 80:1184-1193, 2018.
>
> [R3] Yucen Lily Li et. al. "A Study of Bayesian Neural Network Surrogates for Bayesian Optimization" ICLR 2024.

---

### Official Review · Reviewer_uTQN · 2024-11-02

**Soundness:** 2
**Presentation:** 2
**Contribution:** 2
**Rating:** 3
**Confidence:** 3

**Summary:**

This paper proposes a Bayesian neural network-based semi-supervised learning framework for tackling constrained optimization problems. The authors target engineering problems where labelled data and computational resources are limited. The proposed approach alternates between a supervised learning step which minimizes the cost of the optimization problem and an unsupervised learning step which enforces the related constraints. The authors claim that their approach outperforms conventional methods while significantly reducing equality and feasibility gaps.

**Strengths:**

- The paper is relatively well-organised and easy to understand;
- The paper offers a fairly well-executed application/combination of Bayesian semi-supervised learning techniques to a realistic real-world problem in constrained optimization;
- The Sandwich BNN framework idea is fairly interesting and seems to have direct applicability to general time-sensitive and resource-limited engineering optimization problems

**Weaknesses:**

- As far as I can tell, the work is primarily applied and there is limited technical novelty, which could reduce the community's interest in this paper;
- The paper emphasizes the benefits of Bayesian neural networks for uncertainty estimation but offers no uncertainty calibration analysis or comparisons with other viable uncertainty estimation methods. It is important to note that Bayesian neural networks tend to underestimate variance (partly) due to the mean-field assumption;
- The assumption that the problem has a feasible solution strikes me as quite strong in the general case, for example in complex highly non-convex settings. I would be interested in hearing the author's thoughts on this;
- There is very limited discussion about the drawbacks and/or limitations of the approach.

In my view, the paper could be improved by broadening comparisons with other Bayesian and non-Bayesian uncertainty methods and clarifying and testing the feasibility assumptions. The apparent lack of significant innovation could reduce the impact of the paper, as similar methods (e.g., ensembles or conventional semi-supervised frameworks) could potentially achieve comparable results without the additional complexity of Bayesian inference.

**Questions:**

See above

---

> ### Author Response · Authors · 2024-11-23
> **Reply to Reviewer uTQN**
>
> We thank the reviewer for their comments.
>
> Weakness:
>
> 1. The authors would kindly like to understand the reviewer's perspective on why the proposed work, which introduces a novel semi-supervised learning approach to solve constrained optimization problems in practically significant limited data and time settings, is perceived as an applied work with limited technical novelty.
>
> 2. Authors would like to know what uncertainty calibration analysis the reviewer is referring to. Figure 3 and subsequent results within the paper provide extensive evidence on the effectiveness of proposed method in providing variance information and variance in error. We have also used a multiplier (2 specifically) to avoid the issues for underestimation of variance, if any, due to the mean-field assumption in variational Bayesian inference.
>
> 3. We would like to argue that it is standard practice to assume that at least one feasible solution exists for every input when studying machine learning models for constraint optimization problems. All previous works on ML proxies including DC3, Park & Pascal, Zumzum & Kyri etc. work under this assumption. Even in practice, this is not a limiting assumption for ACOPF as power system is operating at a feasible solution at every instance.
>
> 4. A limitation of this work stems from its motivation that it is targeted to a specific setting of limited training data and time, and does not provide a general method which minimizes violation in general ACOPF proxy settings. Besides, a well known limitation is the high training time requirement by BNN, compared to DNNs. We will update the manuscript in the revision, explicitly mentioning these. We do however believe that despite its higher training complexity BNN is the better choice in the limited data and time setting.
>
>
> We would like to highlight that the paper already presents a lot of comparisons with existing methods, especially the ones that are relevant to the power systems setting.  Additionally, we will update the manuscript with additional discussion and results, which can be found on this link: https://drive.google.com/file/d/1Y57JVPegi2HY2krnLb7qihMvQNdWvfdg/view?usp=share_link
>
> Again, we would like to emphasize that methods like ensembles will require a lot of training time, which is beyond the problem setting of this paper. We will include additional discussion in the introduction to clarify this point.

---

### Official Review · Reviewer_Vowh · 2024-11-03

**Soundness:** 2
**Presentation:** 3
**Contribution:** 1
**Rating:** 5
**Confidence:** 4

**Summary:**

The paper proposes Bayesian Neural Networks (BNNs) for solving the OPF problem under uncertain demand. It employs a novel semi supervised training procedure that reduces the dependence of the training to labelled input-output pairs. The study also suggests using Bernstein bound with Mean Predictive Variance to assess the BNN model performance on out-of-sample inputs.

The authors consider two BNN training methods: a supervised approach using labelled data and a semi-supervised approach, or sandwich learning, which alternates between supervised and unsupervised loss functions iteratively. The latter enforces feasibility by utilizing a function that takes the value 0 if the constraints are satisfied.

The model was applied to 57-, 118-, 500- and 2000-bus test cases and compared against 5 alternative methods in the literature.

**Strengths:**

1.	The main advantage is that the BNN seem to have some performance advantages over DNNs for small and large systems even with training datasets as small as 512 observations. While the optimality gaps are close among all methods, there are notable power balance gap differences between BNN and DNN methods.
2.	In case of large systems such as 2000-bus system, with 512 observations, while DNN training fails, all BNN models still able to perform well. This is particularly useful when it is computationally expensive to generate datasets.
3.	Another contribution is using unsupervised loss term in the training loop, which can work like regularization term, focusing the attention of training to model parameters that would generate more realistic (feasible) solutions. The sandwich training that alternates between supervised and unsupervised data is akin to Physics Informed Neural Network training proposed by [1]. While PINN training contains all KKT conditions in one loss function, the proposed Sandwich learning alternates between two loss functions.

[1] Nellikkath, Rahul, and Spyros Chatzivasileiadis. "Physics-informed neural networks for minimising worst-case violations in dc optimal power flow." 2021 IEEE International Conference on Communications, Control, and Computing Technologies for Smart Grids (SmartGridComm). IEEE, 2021.

**Weaknesses:**

1.	The optimality gaps among different methods are very close for 118- and 500-bus cases. The only exception is 57-bus system where sandwich learning shows some improvement. It looks the only consistent advantage is less power balance violations.
2.	It is not clear if sandwich learning improves BNN model performance. The differences among proposed BNN approaches seem marginal and is not consistent among all test cases.
3.	If I understand correctly, the DNNs in the study are also trained with 512 data points and the training was terminated after 600 seconds (please correct me at this point if I missed the relevant part of the script). My main concern is that training DNNs (with two hidden layers and n_hidden = 2 x input size) with 512 observations is not fair. DNNs are universal function approximators with number of linear facets are determined by hidden layer depth and width [2] and the training must be carried out with dense enough datasets for a good estimation especially in case of larger systems. I understand this is one of the advantages of using BNNs but larger training datasets can still be feasible to construct.
4.	It is not surprising that in case of 2000-bus system, the training would fail with a small dataset and short training time because the NN model (with hidden layer width of 2 x input size) has too many trainable parameters. The authors could find the required training size for DNNs to perform as well as BNNs as a proof of scalability of the latter.

[2] Montufar, G. F., Pascanu, R., Cho, K., & Bengio, Y. (2014). On the number of linear regions of deep neural networks. Advances in neural information processing systems, 27.

**Questions:**

1.	How many data points were used to train the DNN models?
2.	I wonder if the authors tried to use Sandwich learning to train DNNs. As it reduces the dependence on labelled datasets, it can be a good improvement to scale up to larger systems with limited observations.
3.	Some papers in the literature call a shallow NN a model with two hidden layers i.e. input -> hidden and hidden -> output. Others refer shallow NNs as one layered models. If it is the case of the former, to name it as DNN it must have more than two layers. I wonder which is the case in this study. If a shallow NN was trained, then it deeper models can have better advantages.

---

> ### Author Response · Authors · 2024-11-23
> **Reply to Reviewer Vowh**
>
> We thank the reviewer for their thoughtful comments.
> 1. Yes, our primary focus was on feasibility without compromising on optimality, in learning time and data constrained settings. The primary motivation of this paper is to develop an Optimization proxy under a practical but challenging situation where total labeled data for training as well as total time for training and testing are constrained, as shown in Figure 1 on this link: https://drive.google.com/file/d/1Y57JVPegi2HY2krnLb7qihMvQNdWvfdg/view?usp=share_link
> This is different from situations where a large number of training samples can be generated or much longer training time $T_{training}$ for model tuning and testing time $T_{prediction}$ for prediction are available. Technically the number of training data is itself a proxy for data generation time $T_{Data}$. In this context, note that $T_{prediction}$ for our model is at least **10 times lower** than the DC3 approach or Zumzum & Baker's approach because our model does not involve a projection step that requires solving power flows or access to a power flow solver during either training or testing. When compared to models that have fast testing times, our results show up to an order of magnitude improvement in feasibility gaps, without compromising the optimality of predictions. E.g.: the optimality gap reduces to 0.089 by proposed method from 1.284 best point prediction method for case118 in Table 2. Second, our proxy, due to the use of BNN, is able to generate non-trivial confidence bounds on the accuracy of prediction despite a limited number of 1000 validation data. As shown in Figure 4, prior DNN based models when using the same validation dataset, and Hoeffding's or Empirical Bernstein inequality, do not produce meaningful performance bounds.
>
> 2. We emphasize that our contributions extend beyond sandwich learning BNNs to include simple BNNs for constrained optimization and Bernstein inequality-based error guarantees. The results demonstrate that BNNs are well-suited for data- and time-constrained environments. For smaller power grids, the sandwich model effectively reduces power balance violations, achieving its intended goal. However, for larger systems, limited training time prevents the model from fully utilizing available information. Nonetheless, to maintain practicality, we uniformly restrict training to 10 minutes for all examples.
>
> 3. Yes, the DNN training used 512 samples and was capped at 600 seconds. The goal was to compare BNN and DNN performance under low data and limited training time conditions. While larger datasets could be constructed, the time required increases significantly with system size, as noted in [Peng-Pascal paper]. More critically, this added data creation time is a major drawback in power system operations, where datasets are often unavailable or difficult to obtain, especially for N-1 line outage scenarios. This setting aligns closely with the motivation of the proposed work.
>
> 4. Yes, the DNN model for 2000-bus has too many trainable parameters and is failing because of the very limited data and training time. Also, we agree that with more data, and training time, DNNs will start to perform better and at some point exceed the BNN performance (since DNNs can be trained more efficiently compared to BNN). However, we want to again highlight that our motivation is not to say that BNNs are better than DNNs in general. Our limited submission assertion is that-- under data and time constrained settings BNN based models outperform the DNN models for ACOPF problem, as described in motivation before.
>
> Questions:
>
> 1. We used 512 labeled data points for DNN training across all cases, as outlined in Section 5.
>
> 2. While this work focuses on BNNs we agree that sandwich learning has potential applicability for DNNs, as it could reduce reliance on labeled datasets. However, we wish to emphasize that using a DNN alone can only provide Hoeffding’s and Empirical Bernstein bounds on the expected error, as discussed in Section 4 of the paper. An intriguing direction, without losing the Bayesian nature of predictions, would be to use a DNN exclusively for the unsupervised part of training. However, this approach would require substantial effort in designing sequential priors for the supervised BNN stages that follow the unsupervised DNN stages. We plan to explore this direction in future work.
>
> 3. It is standard practice to train NNs with two hidden layers for the ACOPF problem. Almost all existing works adopt a two-hidden-layer architecture. An intuitive explanation for this stems from the fact that power flow equations are quadratic in nature, so two or three nonlinear transformations should suffice to capture the input-output relationship. While it cannot be ruled out that deeper networks might offer better performance, they are much harder to train within the constrained training time setting of the proposed method due to the significantly larger number of parameters.

---

### Meta-Review · Area_Chair_UZnP · 2024-12-20

**Metareview:**

**Summary**


The paper introduces a semi-supervised Bayesian Neural Network (BNN) framework designed to tackle constrained optimization problems, particularly the  optimal power flow (OPF) problem with uncertain demands and limited labeled data. The approach alternates between supervised learning, which minimizes the optimization problem's cost, and unsupervised learning, which enforces related constraints, referred to as sandwich learning. This method significantly reduces feasibility and optimality gaps by employing a novel training procedure that decreases reliance on labeled data. The authors also implement a Selection via Posterior (SvP) scheme that uses BNN uncertainty estimates to enhance model performance and ensure feasibility. Experimental results across various test cases, including 57-, 118-, 500-, and 2000-bus networks, demonstrate that this BNN framework outperforms traditional deep neural networks (DNNs) and other conventional methods, especially in scenarios characterized by sparse data and computational resource constraints.

**Strengths**

The reviewers unanimously highlighted several strengths of the proposed framework:
* The paper is well-organised and easy to follow.
* The use of BNN seems to have some performance advantages over DNNs for small and large systems even with training datasets as small as 512 observations.
* The proposed unsupervised loss term in the training loop, and the sandwich training scheme used in the paper are interesting and add practical value to the proposed solution.


**Weaknesses**

Several core weaknesses was brought up by the reviewers. These include:
* The paper's representation could be enhanced.
* The lack of comparison with more recent relative baselines, including self-supervised and primal-dual learning approaches due to their higher training times and computational demands. The paper could benefit from reporting such baselines and their corresponding training times.
* While this work focuses on addressing the challenges of limited labeled data and restricted model training times, it is not clearly explained how Bayesian Neural Networks (BNNs) specifically contribute to overcoming these challenges.
* Limited discussions on limitations of the proposed solution.

**Conclusion**

The majority of reviewers recognize the importance of the problem addressed by the paper but criticize the experimental setup and the paper's inadequate positioning, including a lack of rigorous comparisons with recent baselines. Despite a substantial rebuttal from the authors, this did not alter the predominantly negative perception held by the reviewers. I agree with the reviewers and vote for rejection of the paper.

**Additional Comments On Reviewer Discussion:**

Despite my efforts to engage the reviewers in a discussion during the review period to reach a consensus on the paper’s merits and shortcomings, there was no participation in any discussion. However, given the less polarized evaluations of this paper, the discussion was not as critical for this work.

---

### Decision · Program_Chairs · 2025-01-22

Reject